# Multifractal scaling analyses of urban street network structure: The cases of twelve megacities in China

**Yuqing Long, Yanguang Chen** *

Department of Geography, College of Urban and Environmental Sciences, Peking University, Beijing, P.R. China

* chenyg@pku.edu.cn

**Data Availability Statement:** All relevant data are within the paper manuscript and its Supporting information files.

**Funding:** National Natural Science Foundation of China, Grant number: 41671167. The project title is

## Abstract

Traffic networks have been proved to be fractal systems. However, previous studies mainly focused on monofractal networks, while complex systems are of multifractal structure. This paper is devoted to exploring the general regularities of multifractal scaling processes in the street network of 12 Chinese cities. The city clustering algorithm is employed to identify urban boundaries for defining comparable study areas; box-counting method and the direct determination method are utilized to extract spatial data; the least squares calculation is employed to estimate the global and local multifractal parameters. The results showed multifractal structure of urban street networks. The global multifractal dimension spectrums are inverse S-shaped curves, while the local singularity spectrums are asymmetric unimodal curves. If the moment order $q$ approaches negative infinity, the generalized correlation dimension will seriously exceed the embedding space dimension 2, and the local fractal dimension curve displays an abnormal decrease for most cities. The scaling relation of local fractal dimension gradually breaks if the $q$ value is too high, but the different levels of the network always keep the scaling reflecting singularity exponent. The main conclusions are as follows. First, urban street networks follow multifractal scaling law, and scaling precedes local fractal structure. Second, the patterns of traffic networks take on characteristics of spatial concentration, but they also show the implied trend of spatial deconcentration. Third, the development space of central area and network intensive areas is limited, while the fringe zone and network sparse areas show the phenomenon of disordered evolution. This work may be revealing for understanding and further research on complex spatial networks by using multifractal theory.

## 1 Introduction

Scientific research includes two processes: one is description, and the other is understanding. Science should proceed first by describing how a system and its parts work and then by understanding why [1, 2]. The precise description relies heavily on mathematics and measurements [2]. Conventional mathematical modeling and quantitative analysis are based on

Modeling geographical systems of cities using spatial correlation functions. The funders had no role in study design, data collection and analysis, decision to publish, or preparation of the manuscript.

**Competing interests:** The authors have declared that no competing interests exist.

characteristic scales [3–6]. Unfortunately, complex systems such as cities bear no characteristic scale. In this case, the concept of characteristic scale should be replaced by the idea of scaling. Traffic networks proved to be typical complex spatial systems with no characteristic scale [7–12]. Fractal geometry provides a powerful tool for describing scale-free phenomena and can be used to make scaling analysis of traffic networks [13–16]. Previous studies have demonstrated that traffic or transport networks, including railways, roads, and urban streets, bear fractal properties, and can be characterized by using fractal dimension [17–22]. In fact, the fractal research on traffic networks can be traced back to the 1960s, when Smeed found that the density distribution of urban street and road network from center to periphery follows inverse power function [23]. The scaling exponent of Smeed's distribution is a function of fractal dimension [13, 18]. This dimension can be termed radial dimension [24, 25]. The radial dimension proved to be a special spatial correlation dimension of fractals [12]. In short, fractal dimension can serve as a spatial characteristic quantity of complex traffic networks.

In literature, fractal studies on cities and traffic networks fall into two categories: monofractal analyses and multifractal analyses. Most studies have focused on the monofractal properties, under the assumption that each pattern can be characterized by a single scaling process. Nevertheless, with regard to urban system, it has been extensively accepted that a single fractal dimension is not enough to depict its complex nature due to spatial heterogeneity [26]. Based on more than one scaling process, multifractal analysis is required. Multifractal scaling is an effective approach to characterize various heterogeneous phenomena in nature and society [27, 28]. It provides a series of parameter spectrums adequately capturing the spatial heterogeneity of fractal patterns and the statistical distribution of measurements across a series of spatial scales [29]. With the help of multifractal modeling, we can study urban systems and traffic networks from different angles and levels. Multifractal theory has been applied to human geography for a long time [29–38]. However, there are few reports on multifractal research of traffic networks. Hierarchy and network structure represent two different sides of the same coin [14]. In many cases, it is hard to model network structure mathematically, but it is relatively easy to model hierarchical structure using proper mathematical tools. In this regard, multifractal theory may provide an advisable approach to studying self-organized complex networks such as urban transportation through self-similar hierarchical networks. To date, few studies have been able to draw on any systematic research into the multifractal structure of traffic networks.

Complex system evolution is mainly based on multifractal scaling processes rather than monofractal scaling process. Now, there is no doubt that the traffic networks have statistical self-similar structure. However, it is not very clear whether the traffic networks in China are of monofractal structure or multifractal structure. Before deeply studying the complexity of traffic networks, we must first find out whether the traffic networks generally have multifractal structure. If so, what are the common characteristics of different traffic networks? Only by properly describing the general multiscaling characteristics of traffic networks, can we expect to further research the evolution mechanism behind them. Therefore, this paper is devoted to exploring the general regularities of multifractal structure of urban traffic networks represented by street links. The aim is at providing some empirical foundations for further research on internal mechanisms and general principles of complex spatial networks. The street networks of twelve typical megacities of China are taken as examples. Chinese digital navigation map in 2016 is collected as materials, and the functional box-counting method is applied to calculate multifractal parameters. The remainder of this article is organized as follows. In Section 2, the multifractal method and two sets of parameters are explained and illustrated. In

Section 3, the definition of study area, data processing methods, and the main results of multi-fractal analysis are displayed. In Section 4, the key points of analyzed results are outlined, and related questions are discussed. Finally, the discussion is concluded by summarizing the main inferences of this work.

## 2 Multifractal models and measurements

### 2.1 Monofractal and multifractals in urban studies

Monofractal is also termed unifractal, which represents simple self-similar structure. The simple fractal model has been widely applied to urban form and growth [13, 14, 39, 40]. A monofractal system is a self-similar hierarchy with symmetric cascade structure and single scaling process. In a monofractal object, different fractal units bear the same form and dimension value. In contrast, a multifractal system is a self-similar hierarchy with asymmetric cascade structure and multi-scaling processes. In a multifractal object, different fractal units possess different forms and different local fractal dimension values. Where the spatial structure is concerned, monofractals are treated as homogeneous fractals, while multifractals are regarded as heterogeneous fractals. A monofractal can be characterized by a single fractal parameter, while multifractals should be characterized by a series of fractal parameters. Generally, two sets of fractal parameters are employed to characterize multifractals, including global and local parameters. The global parameters include *generalized correlation dimension*, $D_q$, and *mass exponent*, $\tau(q)$, and the local parameters comprise *singularity exponent*, $\alpha(q)$, and *local fractal dimension* of the fractal subsets, $f(\alpha)$. In practice, spatial analysis of multifractal systems are based on multifractal parameter spectrums, including global multifractal spectrum, i.e., $D_q$-$q$ spectrum, and local parameter spectrum, i.e., $f(\alpha)$-$\alpha$ spectrum. The latter is also termed $f(\alpha)$ curve and represents the basic multifractal spectrum.

In the real geographical world, there are seldom monofractal phenomena. Urban form and systems of cities proved to be multifractals [26, 28, 34–37]. The monofractal method can be used to describe the basic characteristics of multifractals, but the multifractal spectrums cannot be employed to describe monofractal structure. Is a traffic network monofractal or multifractals? This can be identified by multifractal spectrums. If a traffic network is of multifractal structure, its global parameter spectrum will be an inverse S-shaped curve, and the local parameter spectrum will be a unimodal curve. On the contrary, if the traffic network is of monofractal structure, its global parameter spectrum will be a horizontal straight line, and the local parameter spectrum will be a point [41]. In the generalized correlation dimension set, there are three basic parameters, that is, capacity dimension $D_0$, information dimension $D_1$, and correlation dimension (in a narrow sense) $D_2$. If $D_0 > D_1 > D_2$ significantly, the global dimension spectrum will be an inverse S-shaped curve. Therefore, the inequality $D_0 > D_1 > D_2$ can display multifractal structure of a traffic network. In contrast, if $D_0 \approx D_1 \approx D_2$, a traffic network can be treated as monofractal structure. This is the simplest approach to distinguishing monofractal from multifractals.

### 2.2 Global multifractal parameters

Global parameters describe the fractal object from an overall perspective and macro level. The generalized correlation dimension $D_q$ is based on Renyi's entropy. It is always expressed as [3,

42, 43]:

$$D_q = -\lim_{\varepsilon \to 0} \frac{M_q(\varepsilon)}{\ln \varepsilon} = \begin{cases} \dfrac{1}{q-1} \lim_{\varepsilon \to 0} \dfrac{\ln \sum_{i=1}^{N(\varepsilon)} P_i(\varepsilon)^q}{\ln \varepsilon}, & (q \neq 1) \\[2em] \lim_{\varepsilon \to 0} \dfrac{\sum_{i=1}^{N(\varepsilon)} P_i(\varepsilon) \ln P_i(\varepsilon)}{\ln \varepsilon}, & (q = 1) \end{cases}, \qquad (1)$$

where $q$ refers to the moment order (-∞<$q$<∞), $M_q(\varepsilon)$ to the Renyi's entrop$y$ with a linear scale $\varepsilon$. Eq (1) suggests that multifractal parameters are actually defined by growth probability $P_i(\varepsilon)$ and corresponding spatial scale $\varepsilon$. The spatial scale indicates size and shape while probability indicates chance and dimension. When measured by box-counting method, $N(\varepsilon)$ refers to the number of nonempty boxes with linear size of $\varepsilon$, and $P_i(\varepsilon)$ represents the growth probability, indicating the ratio of measurement results of fractal subset appearing in the $i$th box $L_i(\varepsilon)$ to that of whole fractal copies $L(\varepsilon)$; that is, $P_i(\varepsilon) = L_i(\varepsilon) / L(\varepsilon)$. At a specific scale $\varepsilon$, the larger $P_i(\varepsilon)$ is, the higher growth probability it has, corresponding to higher density of this fractal subset. With the intensive measure, by changing the value of $q$, attention can be focused on locations with high density ($q \to \infty$), or, conversely, on locations with low density ($q \to -\infty$). Another global parameter, mass exponent $\tau(q)$ can be estimated by $D_q$ value [3, 44]:

$$\tau(q) = (q-1)D_q, \qquad (2)$$

which reflects the scaling property from the viewpoint of generalized mass.

As indicated above, there are three basic parameters in the set of generalized correlation dimension. In Eq (1), the common fractal parameters can be derived. For $q = 0$, $D_0$ refers to the *capacity dimension*; for $q = 1$, $D_1$ refers to the *information dimension*; for $q = 2$, $D_2$ refers to the *correlation dimension* [36]. The capacity dimension reflects space-filling degree, the information dimension indicates spatial uniformity, and the correlation dimension suggests spatial dependence extent. The geographical meanings of the three parameters for traffic networks can be tabulated as below (Table 1). Using capacity dimension, we can describe the development extent of a traffic network; using information dimension, we can show the spatial heterogeneity of a traffic network; using correlation dimension, we can characterize the spatial complexity of a traffic network. The correlation dimension is a measurement of spatial dependence. Spatial correlation suggests that the whole is not equal to the sum of parts, and thus there are nonlinear relationships in a system. Nonlinearity suggests complexity. In this sense,

**Table 1. Geographical meanings of capacity dimension, information dimension, and correlation dimension for traffic network.**

| Parameter | Basic measurement | Spatial meaning |
|---|---|---|
| **Capacity dimension $D_0$** | Space-filling degree | Whether or not a place (box) bears elements of networks |
| **Information dimension $D_1$** | Degree of spatial uniformity | How many network elements appear at/in a place (box) |
| **Correlation dimension $D_2$** | Degree of spatial dependence | If a place bears network elements, how many other network elements can be found within a certain distance from the place (box) |

**Note**: The degree of spatial uniformity and spatial difference represents the two different sides of the same coin, and the spatial difference indicates spatial heterogeneity. The degree of spatial dependence suggests spatial complexity.

the spatial correlation dimension is a quantitative criterion of spatial complexity. However, the powerful function of multifractal analysis lies in the spectral curves, not in a single parameter. The relation between the moment order $q$ and generalized correlation dimension gives the global multifractal spectrum, namely, $D_q$-$q$ spectrum, which is an inverse S-shaped curve.

## 2.3 Local multifractal parameters

Local parameters focus on the micro features of different parts and micro levels in multifractals. Given its heterogeneity, a multifractal set has many fractal subsets, and each part corresponds to a power law:

$$P_i(\boldsymbol{\varepsilon}) \propto \boldsymbol{\varepsilon}_i^{\alpha(q)}, \tag{3}$$

where $\varepsilon_i$ refers to the corresponding linear scale of the $i$th box, and $\alpha(q)$ refers to the strength of local singularity. The scaling exponent is also termed as *Lipschitz-Hölder singularity exponent*, suggesting the degree of singular interval measures [3]. Different values of $\alpha$ correspond to different subsets of multifractals, and different regions may share the same value of $\alpha$. Accordingly, the number of fractal subsets with the same $\alpha$ value under the linear $\varepsilon_i$ is given by

$$N(\alpha, \boldsymbol{\varepsilon}_i) \propto \boldsymbol{\varepsilon}_i^{-f(\alpha)}, \tag{4}$$

where $f(\alpha)$ refers to the fractal dimension of the subsets with singularity strength $\alpha$, named as *local fractal dimension*. The higher the local dimension $f(\alpha)$ is, the larger the number of fractal subsets with singularity $\alpha$ get, and *vice versa*. The $\alpha(q)$ and $f(\alpha)$ compose the set of local parameters of the multifractal sets. The relationship between $f(\alpha)$ and $\alpha$ forms the local parameter spectrum, i.e., the singularity spectrum of multifractals. This is a unimodal curve with an apex $(\alpha(0), f(\alpha(0)))$.

There are two basic models of multifractal growth pattern in urban development. One is spatial aggregation or concentration, and the other is spatial diffusion or deconcentration [28]. The different growing types can be distinguished by the height difference of $f(\alpha)$, $\Delta f = f(q \rightarrow +\infty)$-$f(q \rightarrow -\infty)$. For cities with spatial concentration growth pattern, the central regions are denser, and peripheral regions are sparser. While cities of spatial deconcentration pattern are the reverse. In Fig 1, we give a simple representation of how the different growth models of traffic networks are related to the $f(\alpha)$ spectrum. If the $f(\alpha)$ is high on the left tails and low on the right tails ($\Delta f > 0$), and it slants to the left, the fractal growth is dominated by spatial deconcentration. Conversely, when $\Delta f < 0$, and it slants to the right, this suggests the fractal growth of spatial concentration. These local parameters provide detailed information about local differences in the relative intensity of urban street networks.

There are nonlinear relationships between the global parameters and local parameters. The two sets of parameters can be associated with one another by Legendre transform [44–46]. Legendre transform can be expressed as

$$\alpha(q) = \frac{\mathrm{d}\tau(q)}{\mathrm{d}q} = D_q + (q-1)\frac{\mathrm{d}D_q}{\mathrm{d}q}, \tag{5}$$

$$f(\alpha(q)) = q\alpha(q) - \tau(q) = q\alpha(q) - (q-1)D_q. \tag{6}$$

The box-counting method can be employed to estimate global multifractal parameters, and the direct determination method based on normalized rescaled probability measure was used to estimate local multifractal parameters [38, 39]. Lastly, the ordinary least squares (OLS) linear regression was utilized to estimate fractal parameters to obtain practical spectrums [30].

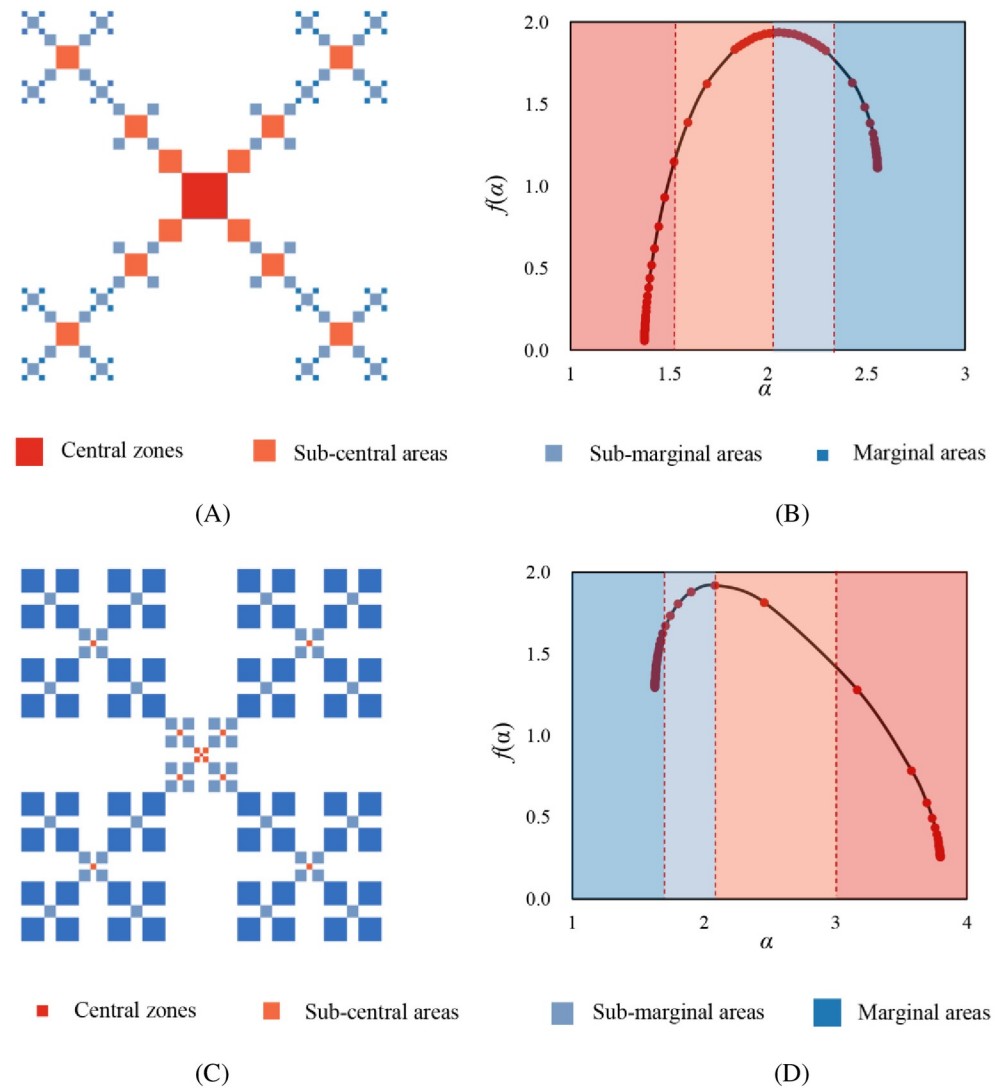

**Fig 1. The singularity spectrums of two different multifractal growth modes: Spatial concentration and deconcentration.** (A) Spatial aggregation mode; (B) Singularity spectrum for spatial aggregation; (C) Spatial diffusion mode; (D) Singularity spectrum for spatial diffusion. **Note**: The squares with larger size represent network intensive areas, while the smaller squares represent network sparser areas. The local parameter $f(\alpha)$ curve stands for the local fractal dimension of the sets of units with the singularity exponent $\alpha$.

In spatial analysis, the global multifractal spectrum can be compared to a telescope, and the local multifractal spectrum can be compared to a microscope. The different levels of a self-similar hierarchical system can be reflected by the moment order $q$. Where the global level is concerned, changing $q$ value indicates changing the described levels of multifractal system; where the local level is concerned, changing $q$ value indicates changing focused parts in the multifractal system. By changing the value of moment order $q$, we can rescale growth probability distribution. One of the advantages of multifractal method is that, by means of multifractal dimension spectrums, different levels of a complex network can be investigated at the macro level, and different parts can be focused at the micro level. Through the $D_q$-$q$ spectrum, we can investigate different levels of a multifractal system on the whole by changing $q$ value. On the other hand, through the $f(\alpha)$-$\alpha(q)$ spectrum, the investigation will focus on parts at different

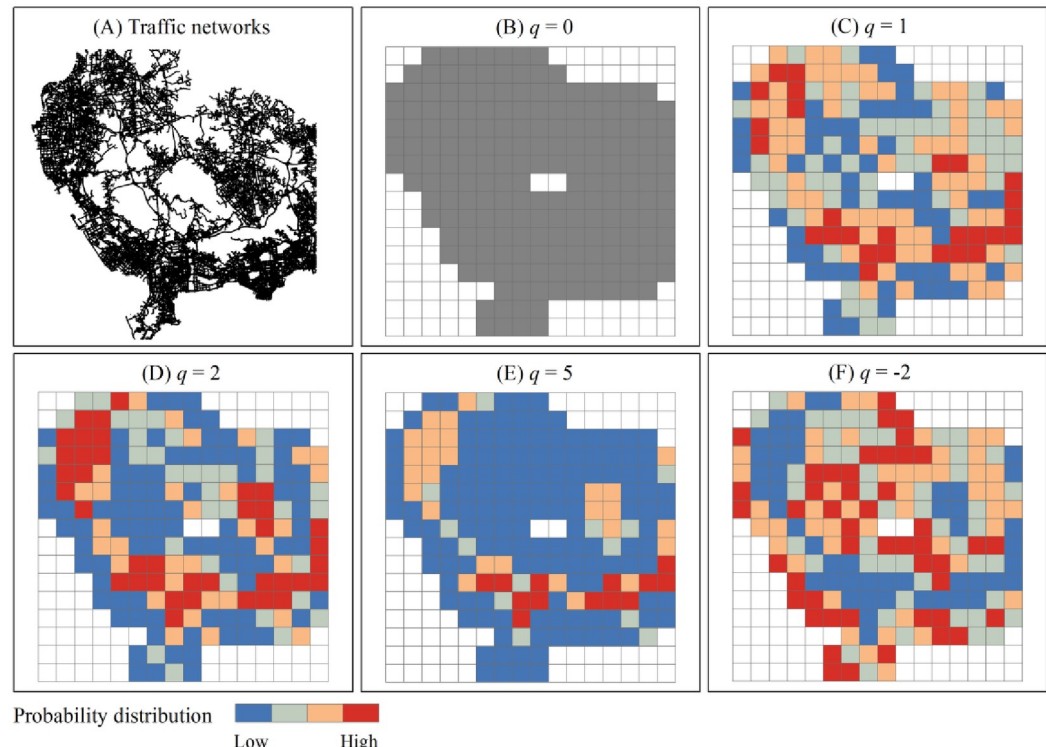

**Fig 2. Schematic representation of rescaling probability distribution for multifractal computation based on box-counting method.** The probability distribution of traffic links is calculated over all boxes and then weighted by moment order $q$. Dark red boxes represent dominant structures with higher weighted probability, and dark blue boxes represent lightweight regions with lower weighted probability. (A) Partial traffic networks in Shenzhen. (B) When $q = 0$, all nonempty boxes are equally weighted (gray). (C) When $q = 1$, the nonempty boxes are weighted by real growth probability. (D)-(E) For $q>0$, the boxes with relative high-density measures gradually gain more importance and their contribution to the entropy will dominate. (F) For $q<0$, the boxes with relatively low-density measures gradually gain more importance and their contribution to the entropy will dominate.

levels of a multifractal system by changing $q$ value. The process of rescaling probability distribution can be illustrated with a simple example of real traffic network (Fig 2).

## 3 Empirical results

### 3.1 Study area, datasets, and methods

The general characteristics of multifractal structure of traffic network can be revealed by induction. In this work, twelve Chinese megacities are selected for case studies of traffic networks. These cities include Beijing, Tianjin, Shanghai, Nanjing, Shenzhen, Guangzhou, Chengdu, Xi'an, Wuhan, Zhengzhou, Shenyang, and Harbin. They are representative cities in typical regions and scattered all over China (Fig 3). The population size characterized by urban permanent residents is near or even greater than five million for each megacity. In these cities, human activities and urban flows are highly concentrated on street networks. Different definitions of cities may affect conclusions regarding the statistical distribution of urban activity [47]. In order to define comparable study areas for different cities so that we obtain comparable multifractal parameters, we first identify the twelve urban boundaries in a consistent way. Several algorithms have been constructed to delimit urban boundaries [48–51]. Among these algorithms, City Clustering Algorithm (CCA) has attracted great attention for its simplicity and efficiency [52, 53]. CCA can be regarded as a method of spatial cluster. In this study, City

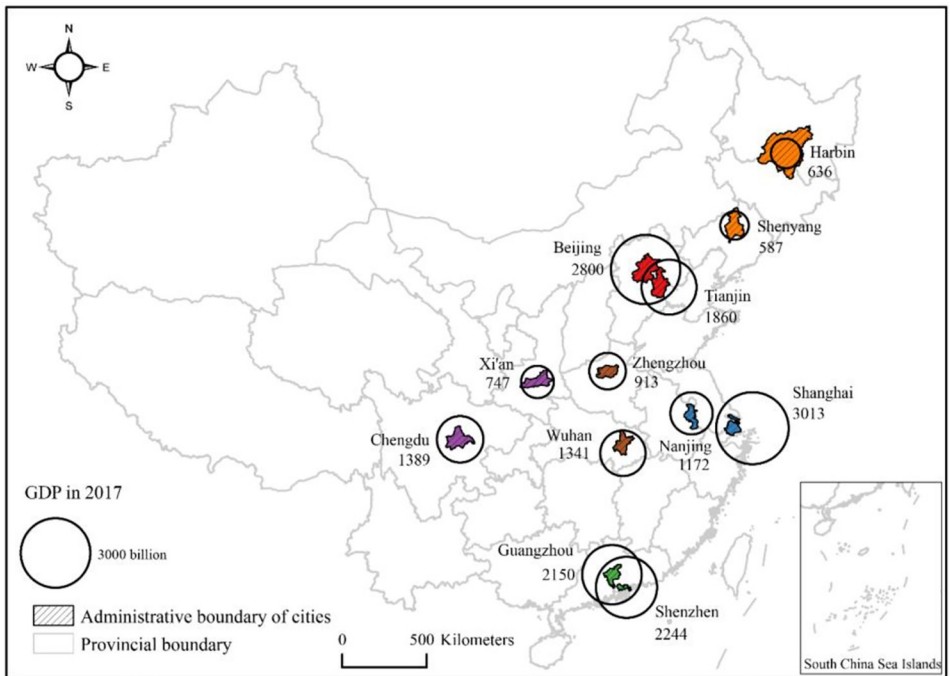

**Fig 3. The spatial distribution of twelve megacities of China.** All of them are the most representative cities in typical regions of China, including North China (Beijing, Tianjin) drawn in red, Northeast China drawn in orange (Harbin, Shenyang), Central China drawn in brown (Zhengzhou, Wuhan), West China drawn in purple (Xi'an, Chengdu), East China drawn in blue (Shanghai, Nanjing), South China drawn in green (Shenzhen, Guangzhou).

Clustering Algorithm (CCA) and head/tail break method are employed to identify urban boundaries and thus define comparable study areas, functional box-counting method is utilized to make spatial measurement for scaling analysis, and regression analysis based on the least squares methods (LSM) is used to estimate multifractal parameters.

The data processing includes three main steps. Step 1: selecting the original data. Street network datasets in 2016 are obtained from the Chinese digital navigation map (http://geodata.pku.edu.cn), including freeways, arterials, and collectors. Using the datasets, we then derive 6.74 million street nodes by ESRI ArcGIS. Step 2: spatial clustering of street nodes. We apply the CCA to cluster contiguous nodes by using the Aggregate Points tool in ArcGIS, which is intrinsically based on a Triangulated Irregular Network (TIN) model. In this process, the aggregation distance is set as 615 meters, which is the mean length of whole TIN edges determined by head/tail breaks [54, 55]. Head/tail breaks is a classification scheme or recursive function for deriving inherent hierarchy or heterogeneity of a dataset [56, 57]. Step 3: defining urban boundaries. We fill the holes inside the clusters and select the urban envelope of the largest cluster of each city to define urban boundary (Fig 4). An urban envelope of a city is a closed boundary curve of an urban area [13, 58]. The comparable study areas of different cities can be defined by urban envelopes based on CCA.

A network is a complicated system comprising a set of line segments (edges) and intersection points (node or vertex). Urban traffic networks consist of streets, roads, and junction points. Multifractal parameter measurement may be based on street links (edges) or street nodes (vertex). This work focuses on multifractal patterns of street links. Because street links may contain more information about local connectivity, which better reflects the network density. As a reference, we also provide the multifractal computation results generated by street

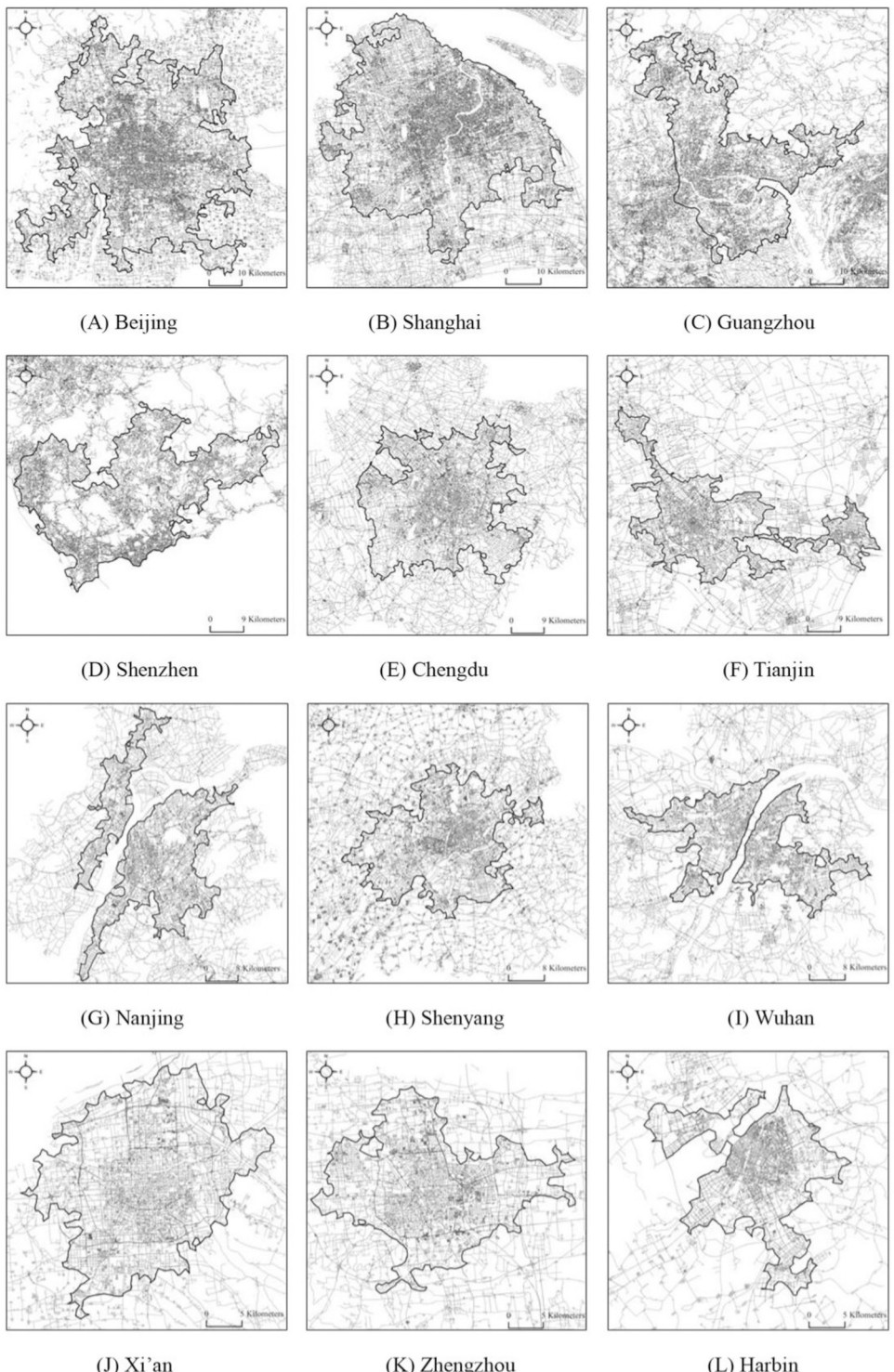

**Fig 4. The distribution of street networks in twelve representative cities.** The black line is the city boundary identified, which is smaller than the municipal area for most cities; the grey lines represent the street network.

nodes (S1 Appendix). The multifractal parameters of the 12 urban street networks can be estimated step by step as follows.

**Step 1: defining the maximum box**. Determine a circumscribed rectangle of urban envelope. The rectangular area is termed *measure area* in mathematics. The measure area can be treated as the maximum box.

**Step 2: extracting the spatial data**. Apply the functional box-counting method to the traffic networks to gain spatial data. The linear size of the largest box is regarded as $\varepsilon = 1$, and the corresponding nonempty box number is $N(\varepsilon) = 1$. The probability is $P(\varepsilon) = 1$. Then the box area is divided into 4 equal parts. The linear size of each part is $\varepsilon = 1/2$. Count the secondary level boxes, and the number of nonempty boxes is $N(\varepsilon) > 1$. Each nonempty box corresponds to a probability $P_i(\varepsilon)$ ($i = 1, 2, \ldots, N(\varepsilon) \leq 4$). Next, the maximum box area is divided into 16 equal parts; that is to say, each secondary level box is divided into 4 equal parts once again. The linear size of each part is $\varepsilon = 1/4$ this time, and the number of nonempty boxes $N(\varepsilon)$ is counted again. Still, each nonempty box corresponds to a probability value $P_i(\varepsilon)$ ($i = 1, 2, \ldots, N(\varepsilon) \leq 16$). The rest can be handled in the same way.

**Step 3: calculating multifractal parameters**. By the least squares regression analysis, the $\mu$-weight method, namely, the direct determination method based on rescaled probability measure can be used to estimate local multifractal parameters (S1 Table). A weighted measure can be defined as

$$\mu_i(\varepsilon) = \frac{P_i(\varepsilon)^q}{\sum\limits_{i=1}^{n} P_i(\varepsilon)^q}. \tag{7}$$

where $\mu(\varepsilon)$ denotes rescaled probability measure based on the given linear size of box $\varepsilon$. Then the singularity exponent $\alpha(q)$ and corresponding local fractal dimension $f(q)$ can be estimated by the following formula [59, 60]

$$\alpha(q) = \lim_{\varepsilon \to 0} \frac{1}{\ln \varepsilon} \sum_{i=1}^{N(\varepsilon)} \mu_i(\varepsilon) \ln P_i(\varepsilon), \tag{8}$$

$$f(q) = \lim_{\varepsilon \to 0} \frac{1}{\ln \varepsilon} \sum_{i=1}^{N(\varepsilon)} \mu_i(\varepsilon) \ln \mu_i(\varepsilon). \tag{9}$$

Then, by Legendre transform, we can obtain the corresponding global multifractal parameters, including the generalized correlation dimension $D_q$ and mass exponent $\tau(q)$. Using Eq (6), we can convert the singularity exponent $\alpha(q)$ and corresponding local fractal dimension $f(q)$ into the generalized correlation dimension $D_q$. Then, using Eq (1) we can convert the generalized correlation dimension $D_q$ into the mass exponent $\tau(q)$. Alternatively, if we firstly compute the global parameter, $D_q$ and $\tau(q)$, by means of Eqs (1) and (2), we can transform them into the local parameter, $\alpha(q)$ and $f(q)$ by using Eq (6) and the discretized form of Eq (5). In this study, the scale range of spatial subdivision is set as $2^0 \sim 2^9$. Besides, the value range from -40 to 40 is selected for $q$, as multifractal parameters are very close to their convergence when $|q|$ approaches 40 (S2 Table). The multifractal parameters are calculated by the OLS linear regression (S3 Table).

**Table 2. Basic fractal parameters of street network for 12 Chinese cities.**

| Region | City | Area $A$ (km$^2$) | Capacity dimension | | Information dimension | | Correlation dimension | |
|---|---|---|---|---|---|---|---|---|
| | | | $D_0$ | $R^2$ | $D_1$ | $R^2$ | $D_2$ | $R^2$ |
| North China | Beijing | 2719 | 1.9106*** | 0.9986 | 1.8642*** | 0.9994 | 1.8339*** | 0.9996 |
| | | | (0.0251) | | (0.0155) | | (0.0126) | |
| | Tianjin | 901 | 1.8409*** | 0.9967 | 1.7997*** | 0.9982 | 1.7726*** | 0.9989 |
| | | | (0.0373) | | (0.0267) | | (0.0207) | |
| South China | Guangzhou | 1660 | 1.8691*** | 0.9985 | 1.8344*** | 0.9993 | 1.8106*** | 0.9993 |
| | | | (0.0252) | | (0.0170) | | (0.01640 | |
| | Shenzhen | 1561 | 1.7825*** | 0.9993 | 1.7474*** | 0.9998 | 1.7228*** | 0.9997 |
| | | | (0.0163) | | (0.0083) | | (0.0097) | |
| East China | Shanghai | 2589 | 1.9064*** | 0.9992 | 1.866*** | 0.9996 | 1.8391*** | 0.9996 |
| | | | (0.0196) | | (0.0137) | | (0.0130) | |
| | Nanjing | 860 | 1.8486*** | 0.9967 | 1.7987*** | 0.9982 | 1.764*** | 0.9988 |
| | | | (0.0376) | | (0.0271) | | (0.0218) | |
| Central China | Wuhan | 633 | 1.8367*** | 0.9964 | 1.79*** | 0.9978 | 1.76*** | 0.9984 |
| | | | (0.0388) | | (0.0297) | | (0.0247) | |
| | Zhengzhou | 489 | 1.8283*** | 0.9958 | 1.791*** | 0.9978 | 1.7632*** | 0.9986 |
| | | | (0.0418) | | (0.0298) | | (0.0231) | |
| West China | Xi'an | 632 | 1.8285*** | 0.9965 | 1.7956*** | 0.9982 | 1.7708*** | 0.9988 |
| | | | (0.0382) | | (0.0269) | | (0.0220) | |
| | Chengdu | 1348 | 1.8804*** | 0.9973 | 1.8307*** | 0.9981 | 1.7974*** | 0.9985 |
| | | | (0.0349) | | (0.0283) | | (0.0247) | |
| Northeast China | Shenyang | 755 | 1.8747*** | 0.9972 | 1.8286*** | 0.998 | 1.7991*** | 0.9985 |
| | | | (0.0352) | | (0.0287) | | (0.0246) | |
| | Harbin | 326 | 1.788*** | 0.9959 | 1.755*** | 0.9986 | 1.7361*** | 0.9992 |
| | | | (0.0407) | | (0.0232) | | (0.0177) | |

**Note**: The robust Standard Errors are quoted in parenthesis.

*** significant at 1%.

## 3.2 Global multifractal spectrums

First of all, let's examine whether or not the traffic networks bear multifractal properties. The simplest way is to compare the three basic fractal parameters in the generalized correlation dimension set: *capacity dimension* $D_0$, *information dimension* $D_1$, and *correlation dimension* $D_2$ (Table 2). Empirical results show that the street network bears multifractal structure in Chinese cities. Apparently, $D_0 > D_1 > D_2$ holds for all the 12 cities. The three basic multifractal parameters reveal useful information about the overall spatial coverage and dependence of urban street networks. In general, the space-filling degree, spatial equilibrium degree, and spatial dependence degree of urban street networks for each city are generally high. The $D_0$, $D_1$, and $D_2$ are more than 1.7. Statistical analyses show that there is a significant correlation between each fractal dimension and city size. The urban sizes have a close correlation to the fractal dimension values of street networks with an exception of Shenzhen city. If we use the urban area to measure city size, we can find a logarithmic relation such as $D_q = a + b\ln(A)$, where $a$ and $b$ are parameters. The city of Shenzhen is still an outlier. The absolute value of the standardized prediction error of Shenzhen goes beyond 2. If Shenzhen is removed, the goodness of fit ($R^2$) between the urban area logarithm and the fractal parameters $D_0$, $D_1$, and $D_2$ are 0.8789, 0.9724, and 0.9197, respectively. If Shenzhen is taken into account, the values lessen to

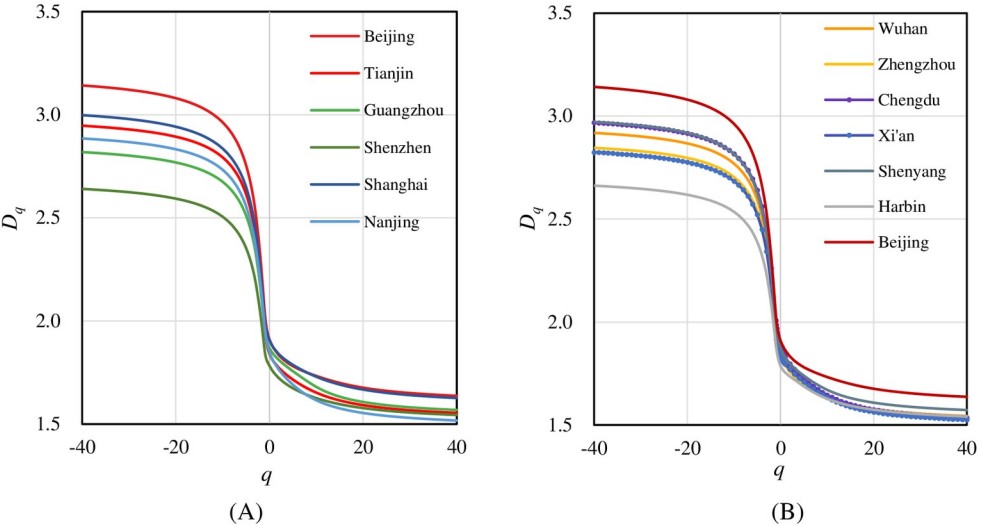

**Fig 5. The generalized correlation dimension spectrums of street networks of 12 Chinese cities.** The $D_q$ curve of Beijing appears in both (A) and (B) as a reference. They are monotonic decreasing functions of $q$, indicating multifractal property. The right tails of $D_q$ curves and convergence values of $q>0$ are very close to each other for most cities. While the left tails of $D_q$ curves exhibit wider gaps. This means that the development of traffic networks in the central areas of different cities is similar, but there are significant differences in the suburbs from city to city.

0.4504, 0.4710, and 0.4666. Using the urban population to replace the urban area, we have a linear relation as follows: $D_q = c+dP$, where $c$ and $d$ are parameters. Shenzhen is still an outlier. Shenzhen is special because there are many natural reserved areas such as large ecological parks throughout the city. In particular, Shenzhen can be seen as a *shock city* in the process of Chinese urbanization. In urban geography, *shock city* is regarded as the embodiment of surprising and disturbing changes in economic, social, and cultural life [61]. Shenzhen's population growth exceeds the development of traffic network. So the $D_0$, $D_1$, and $D_2$ values of Shenzhen's street network turn out to be lower relative to its city size.

Further, we check the generalized correlation dimension spectrums. The spectral lines of $D_q$ changing over $q$ are all inverse S-shaped curves (Fig 5). In generalized correlation dimension spectrums, the macrostructure of street networks in twelve cities shows both similarities and differences. First, all these traffic networks of cities have formed multifractal structure. All the $D_q$ spectrums take on a similar inverse S shape. The value of $D_q$ exceeds the Euclidean dimension of the embedding space $d_E = 2$ when $q<0$, which is abnormal. In theory, based on box-counting methods, the multifractal dimension values should come between topological dimension $d_T = 0$ and embedding dimension $d_E = 2$. Second, the development of traffic networks in the central areas of different cities is similar, but there are significant differences in the suburbs from city to city. The right tails of $D_q$ curves and convergence values of $q>0$ are very close to each other for most cities. This indicates that the street networks in central areas tend to develop a similar structure among cities. In particular, the $D_q$ curves of Beijing and Shanghai are higher in the right tails, showing the more compact structure of street networks in central areas. On the contrary, the left tails of $D_q$ curves exhibit larger gaps, suggesting more dissimilarities among cities are mainly observed in urban fringe and sparse areas.

### 3.3 Local multifractal spectrums

Local fractal parameters and multifractal spectrums bring the local and micro features into focus. As for the singularity exponents $\alpha(q)$, the scaling relationships maintain well at different

**Table 3. Multifractal parameters of street network for each city by OLS regression method.**

| Region | City | $D_{-40}$ | $D_{+40}$ | $\alpha_{-40}$ | $A_P$ | $f_{-40}$ | $f_{+40}$ | $\Delta f$ |
|---|---|---|---|---|---|---|---|---|
| **North China** | Beijing | 3.142*** | 1.637*** | 3.2063*** | 1.5978*** | 0.5717** | 0.0717 | -0.5001 |
| | | (0.2603) | (0.0271) | (0.2718) | (0.0283) | (0.2282) | (0.1077) | |
| | Tianjin | 2.9468*** | 1.5541*** | 3.0027*** | 1.5159*** | 0.7112*** | 0.0275 | -0.6837 |
| | | (0.3104) | (0.0228) | (0.3228) | (0.0225) | (0.2029) | (0.0326) | |
| **South China** | Guangzhou | 2.8189*** | 1.5684*** | 2.8717*** | 1.531*** | 0.7078*** | 0.0723 | -0.6355 |
| | | (0.2815) | (0.0376) | (0.2931) | (0.0386) | (0.2070) | (0.0992) | |
| | Shenzhen | 2.6404*** | 1.5442*** | 2.6899*** | 1.5091*** | 0.6593*** | 0.1389 | -0.5204 |
| | | (0.3336) | (0.0164) | (0.3458) | (0.0164) | (0.1738) | (0.0803) | |
| **East China** | Shanghai | 2.998*** | 1.6265*** | 3.0566*** | 1.5856*** | 0.656** | -0.0077 | -0.6637 |
| | | (0.3533) | (0.0279) | (0.3667) | (0.0284) | (0.2087) | (-0.0876) | |
| | Nanjing | 2.8851*** | 1.5174*** | 2.9401*** | 1.4801*** | 0.6872** | 0.0265 | -0.6607 |
| | | (0.2980) | (0.0213) | (0.3103) | (0.0213) | (0.2125) | (0.0547) | |
| **Central China** | Wuhan | 2.9187*** | 1.5438*** | 2.9719*** | 1.5094*** | 0.7913*** | 0.1673* | -0.6240 |
| | | (0.2554) | (0.0250) | (0.2665) | (0.0256) | (0.2199) | (0.0926) | |
| | Zhengzhou | 2.8461*** | 1.538*** | 2.8972*** | 1.5025*** | 0.8011*** | 0.1185 | -0.6826 |
| | | (0.3100) | (0.0176) | (0.3216) | (0.0177) | (0.1734) | (0.0666) | |
| **West China** | Xi'an | 2.8248*** | 1.5257*** | 2.8757*** | 1.4899*** | 0.788*** | 0.0949 | -0.6930 |
| | | (0.3723) | (0.0253) | (0.3858) | (0.0267) | (0.1797) | (0.0954) | |
| | Chengdu | 2.9685*** | 1.5353*** | 3.0229*** | 1.4969*** | 0.7896*** | -0.0010 | -0.7906 |
| | | (0.2392) | (0.0230) | (0.2499) | (0.0233) | (0.2299) | (-0.0830) | |
| **Northeast China** | Shenyang | 2.972*** | 1.5727*** | 3.027*** | 1.537*** | 0.7723*** | 0.1423 | -0.6300 |
| | | (0.2641) | (0.0256) | (0.2752) | (0.0246) | (0.2283) | (0.0905) | |
| | Harbin | 2.6623*** | 1.5411*** | 2.7086*** | 1.5064*** | 0.8104*** | 0.1551*** | -0.6552 |
| | | (0.3160) | (0.0356) | (0.3292) | (0.0352) | (0.2228) | (0.0439) | |

**Note**: The robust Standard Errors are quoted in parenthesis.

*** significant at 1%;

** significant at 5%;

* significant at 10%.

levels with $q$ changes (Table 3). The shape of $\alpha(q)$ curve is similar to the $D_q$ spectrum, taking on an inverse S-shaped curve. But it changes more steeply. According to Eq (8), the spectrum of singularity exponent $\alpha(q)$ is the increment curve of the mass exponent $\tau(q)$ over moment order $q$. When the moment order $q$ tends to be positive or negative infinity, the singularity exponent gradually approaches the generalized correlation dimension. In this sense, for the extreme conditions, the $\alpha(q)$ spectrum gives similar information to the $D_q$ spectrum. Where central areas are concerned, the $\alpha(q)$ spectral lines are similar. While for the urban fringe and sparse areas, there are distinct differences of $\alpha(q)$ spectrums among cities (Fig 6). However, the $\alpha(q)$ spectrums give a critical point of moment order: $q = -2$. If $q<-2$, the singularity exponent shows no significant change over $q$ for all the urban traffic networks. This suggests that, for lower levels and small elements of a traffic network, the hierarchy with cascade structure has not been developed in the study period. Thus, different lower levels of a traffic network have no significant difference.

The basic character of a multifractal system is that different parts bear different fractal dimensions. The local dimension is expressed as $f(q)$. According to Eq (7), $f(q)$ is based on information entropy, which is based on rescaled probability distribution. Correspondingly,

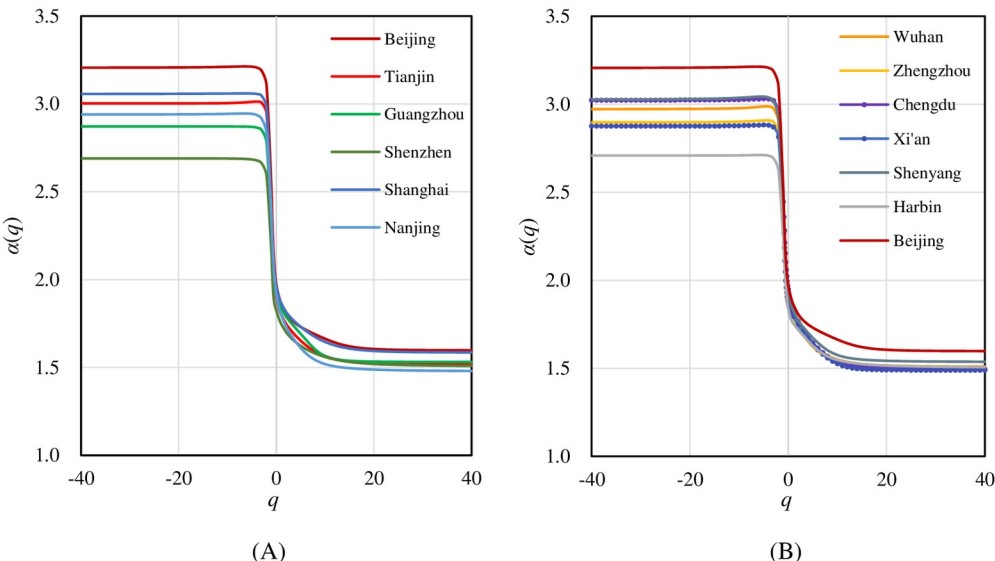

(A)  (B)

**Fig 6. The singularity exponent spectrums of street networks for different cities.** The $\alpha(q)$ curve of Beijing appears in both (A) and (B) as a reference. The singularity exponent $\alpha(q)$ curve is a monotonic decreasing function of $q$. Generally, the left tails of curves exhibit distinct differences among cities.

in terms of Eq (6), $\alpha(q)$ is based on cross entropy. The main spatial information from the $f(\alpha)$ spectrum is as follows. First, multifractal scaling property. The $f(q)$-$q$ curves and $f(\alpha)$- $\alpha$ spectrums are all unimodal curves rather than points (Figs 7 and 8). This lends more support to the judgment that all the traffic networks bear multifractal structure. Second, spatial growth patterns. The $f(\alpha)$ curves suggest that the development of traffic network embodies the spatial concentration trend. The $f(q)$ spectrums take on non-symmetric shape, high on the left and low on the right (Fig 7). Besides, local singularity spectrum $f(\alpha)$-$\alpha$ is shown as a unimodal

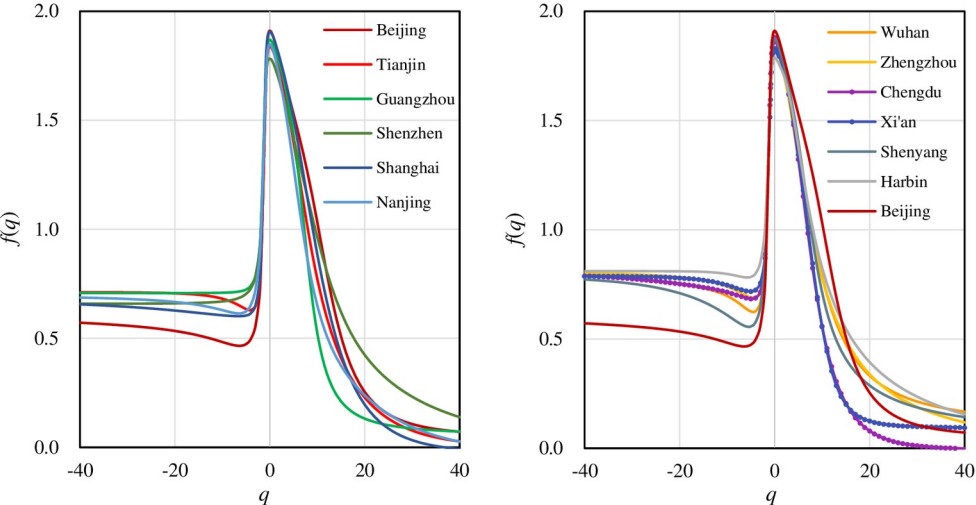

**Fig 7. The local fractal dimension spectrums of street networks for different cities.** The $f(q)$ curve of Beijing appears in both (A) and (B) as a reference. The local fractal dimension $f(q)$ curve is a distinct non-symmetric shape curve, high on the left and low on the right. Besides, the $f(q)$ spectrums of most cities show an abnormal decrease when $q<-2$. Guangzhou and Shenzhen are exceptions.

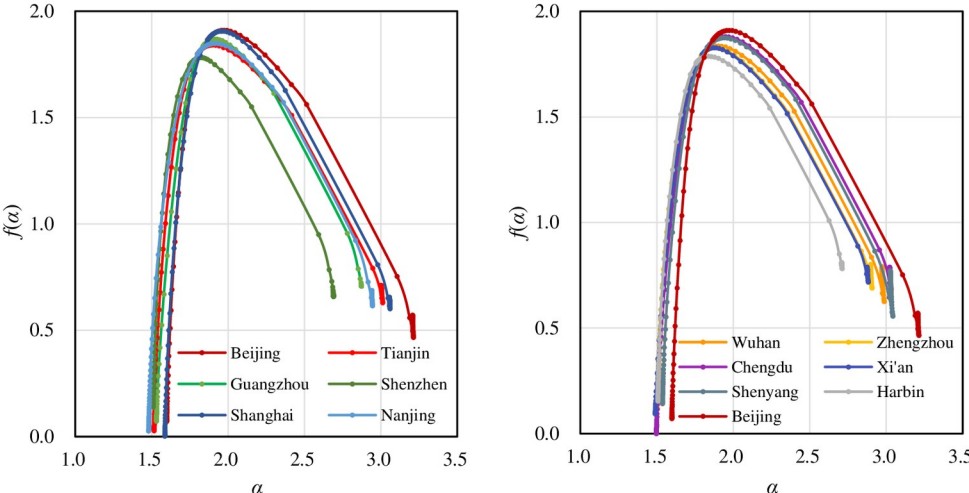

**Fig 8. The local singularity spectrums of street networks for different cities.** The singularity spectrum $f(\alpha)$ is an asymmetric unimodal curve, low on the left tails and high on the right tails. The $f(\alpha)$ spectrum inclines to the left.

curve, low on the left tails and high on the right tails. However, the $f(\alpha)$ curve slants to the left (Fig 8), implying a contradictory trend of spatial deconcentration. Third, spatial development problems. The left side of $f(q)$ spectrums show an abnormal decrease in most cities. In theory, the standard $f(q)$ spectrum is monotonic increasing when $q<0$ and monotonic decreasing when $q>0$ [41], as the green curves in Fig 7A. While in our practical $f(q)$ spectrums, many curves present obvious partial mutation. This implies there are some structural disorders in sparse areas and urban fringes. In contrast, the $f(q)$ curves of Guangzhou and Shenzhen display no mutation. Studies have found that the fractal dimension growth curve of cities in South China is different from that in North China: the former can be described by the ordinary logistic function, and the latter can be described by the quadratic logistic function [62]. Based on these facts, it can be inferred the development of cities in South China is more significantly acted by self-organization and the market economy of bottom-up evolution [35, 62].

The double logarithmic plots (log-log plots) for local multifractal parameter estimation can be employed to display the problem of microstructure of traffic networks. The scaling relations of local dimensions $f(q)$ suggest disordered distribution of street links in two extreme regions: highly dense areas and very sparse areas. With the absolute value of $q$ increases, the goodness of fit of $f(q)$ decreases seriously, and the scattered points in the log-log plots become more and more chaotic correspondingly (Fig 9). This indicates the degradation of multifractal scaling relations. Despite this, as shown in Table 3, the scaling relations of singularity exponent $\alpha(q)$ remain stable as $q$ approaches infinity. Table 4 summarizes the statistic thresholds of the moment order $q$ based on significance level $\alpha = 0.05$ of local dimensions $f(q)$. The value of $q$ reflects different levels of a system. The structural levels vary greatly among cities. Generally speaking, the range of $q>0$ is narrower than that of $q<0$, indicating worse fractal structure in central areas. For Beijing, the fractal relation in the sparse areas is degraded quickly ($q = -4$), and there are poorer levels in high-density areas. Guangzhou and Xi'an show fewer levels in dense areas as well. While Shenzhen and Zhengzhou exhibit rich levels in both sparse areas and dense areas. This suggests that, for the traffic networks in the real world, multifractal structure develops within a limited scaling range.

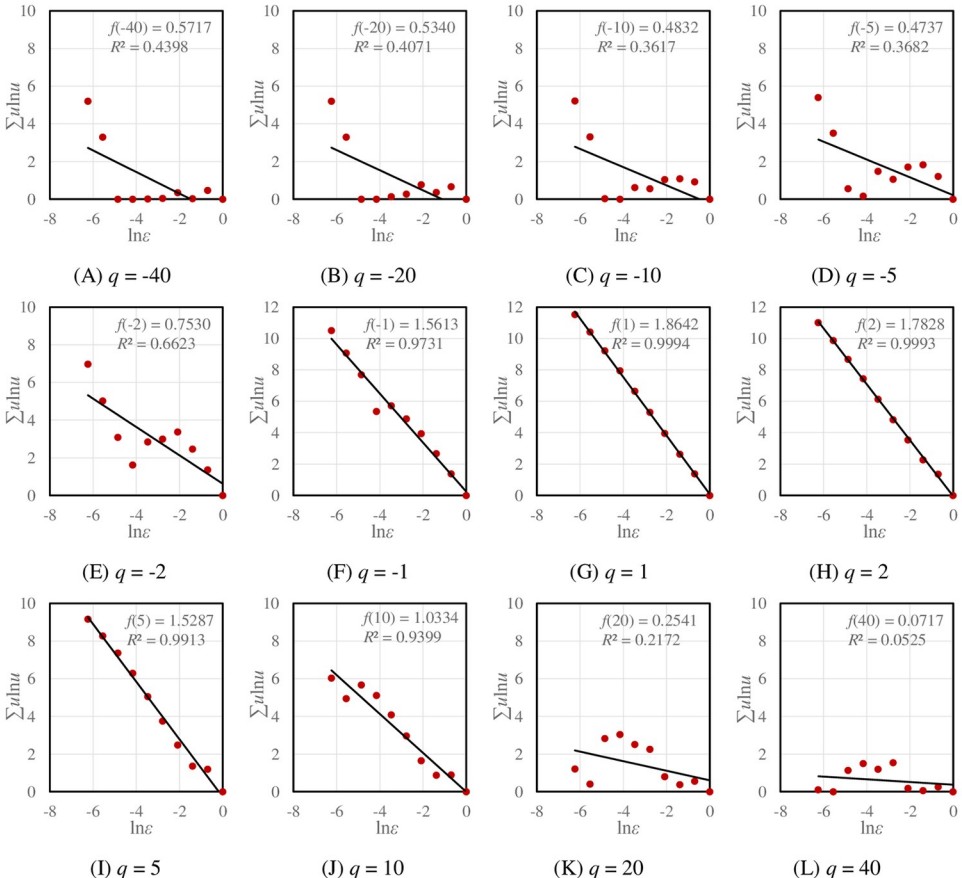

**Fig 9. The log-log plots for estimating the local fractal dimension _f(q)_ of Beijing's traffic network with changes of _q_.** With the absolute value of _q_ increases, the scattered points in log-log plots become more and more disordered, and the scaling relationships are broken seriously.

## 4 Discussion

The above calculations and analyses show that traffic networks of urban streets and roads are complex spatial systems with multiscaling fractal patterns and processes. The general spatial features of traffic networks can be revealed. **Firstly, all these urban traffic networks bear**

**Table 4. The statistic thresholds of the moment order _q_ based on significance level $\alpha = 0.05$.**

| Region | City | For $q<0$ | For $q>0$ | Region | City | For $q<0$ | For $q>0$ |
|---|---|---|---|---|---|---|---|
| **North China** | Beijing | -4 | 16 | Central China | Wuhan | -40 | 26 |
| | Tianjin | -40 | 19 | | Zhengzhou | -40 | 34 |
| **South China** | Guangzhou | -40 | 10 | West China | Xi'an | -40 | 11 |
| | Shenzhen | -40 | 33 | | Chengdu | -40 | 13 |
| **East China** | Shanghai | -40 | 16 | Northeast China | Shenyang | -40 | 27 |
| | Nanjing | -40 | 21 | | Harbin | -40 | 40 |

**Note**: If the _q_ value exceeds the statistical threshold, the confidence level of the local fractal relation will be less than 95%. For example, for Beijing, if $q<-4$ or $q>16$, the local fractal relations are not significant at the level $\alpha = 0.05$.

**multifractal structure rather than monofractal structure**. The global multifractal spectrums of the 12 cities in China are inverse S-shaped curves rather than straight lines, and all the local multifractal spectrums are unimodal curves rather than points. **Secondly, the multifractal scaling relations of all these urban traffic networks are confined within certain spatial and hierarchical ranges**. The traffic networks of 12 Chinese cities take on disordered development in road sparse regions and periphery regions, and the fractal structure partially degenerated in street intensive regions and central areas. This suggests that traffic networks are fractal-like systems instead of real fractal systems, and they can be treated as evolutive fractals rather than natural fractals. **Thirdly, multifractal scaling analyses show the similarities and differences between these urban traffic networks**. The similarities reflect the general regularity of traffic networks of the 12 cities, while the differences reflect the characteristics of each city. This work is devoted to revealing the common properties of urban traffic networks, laying a foundation for deep studies on general principles of complex spatial networks (Table 5).

The important function of multifractal in geographical practice is the diagnosis of spatial problems. The gap between the development goal and the status quo of a system is just the problem to be solved. The process of solving problems is the process of geographic system optimization. The spatial optimization of street networks has long been a major topic of concern [63–66]. Only when the problem is made clear can the problem be solved, so as to achieve the aim of optimizing the geographical space. The spatial pattern of urban street network is a complexity problem, and fractal geometry has long been confirmed as a powerful tool to characterize its complexity and nonlinear dynamics. We agree with Batty [67], who once pointed out: "An integrated theory of how cities evolve, linking urban economics and transportation

**Table 5.  The main results and inferences of calculations and analyses on multifractal traffic networks of 12 Chinese cities.**

| Result and explanation | Inference |
|---|---|
| [**Result**] At the global level, the $D_q$ spectrums take on inverse S-shaped curves; at the local level, the $f(\alpha)$ spectrums take on unimodal curves. [**Explanation**] For monofractals, the $D_q$ spectrums become horizontal straight lines, and each $f(\alpha)$ spectrum condenses into a point. For nonfractals, no power laws, or $D_0 = D_1 = D_2 = d$. | The traffic networks of 12 cities in China are multifractal systems without exception. |
| [**Result**] When $q \to -\infty$, we have $D_q >> d = 2$; when $q \to \infty$, the scaling relations for local dimension are broken. [**Explanation**] When $q \to -\infty$, the multifractal spectrums reflect edge areas or network sparse regions; when $q \to \infty$, the multifractal spectrums mirror on central areas or network intensive regions. | The traffic networks of 12 cities in China take on disordered development in road sparse regions and periphery regions. |
| [**Result**] The scaling relation reflecting $\alpha(q)$ does not degenerate significantly with the change of $q$, but the scaling relation of $f(\alpha)$ partially degenerates when the $q$ value becomes too high or too low. [**Explanation**] The scaling relation degeneration means that the power-law distribution is broken, and the scattered points on the corresponding log-log plot do not take on a straight linear trend. | The traffic networks of 12 cities in China follow scaling law in different parts and levels, but the scaling relation precedes fractal structure. |
| [**Result**] Local fractal dimension spectrums are of asymmetry. The left tails of $f(q)$ curves are higher than right tails, while the left ends of $f(\alpha)$ curves is lower than the right ends. However, the tops of the $f(\alpha)$ curves slope to the left. [**Explanation**] Multifractal development falls into two categories: spatial concentration and spatial deconcentration. | The traffic networks of 12 cities in China take on spatial aggregation patterns, but include spatial diffusion trends. |

behavior to developments in (self-organized) network science, allometric growth, and fractal geometry, is being slowly developed." Fractal structure proved to be a type of spatial order emerging at the edge of chaos [65–70]. Fractal geometry provides an efficient approach for scaling analysis. By using fractal geometry, we can combine urban transportation behavior with urban land use pattern and complex network structure. Hierarchical network includes fractal structure, and scale-free network can be associated with fractals. The commonness of complex network and fractals lies in scaling [69, 71–73].

There have been large amounts of previous works concentrating on monofractal properties. In recent years, multifractal approach has been utilized to characterize transport network of London and Spain [26, 74]. Compared with the previous studies on urban street networks, especially, the fractal street networks, the novelty of this work lies in the following aspects. **First, comprehensive empirical analysis of multifractal scaling in urban street networks**. We selected 12 megacities in China as examples, and then defined the study areas in light of the identical standard so that the results are comparable. Through the calculation results of these cities, we can see the general features of multifractal urban street networks. **Second, global analysis and local analysis of multifractal structure of urban street networks**. We utilize the global parameters to reflect the similarity of different street networks, and use the local parameters to reveal both similarities and differences of these networks among cities. The commonness reflects the law of city development, while the differences may reflect the problems to be solved in urban traffic networks. **Third, visual analysis of scaling evolution of local levels of urban traffic networks**. We make use of log-log scatterplots to show how the scaling relation of local levels of street networks changes over the moment order $q$ values. One of revealing findings is that the local scaling relations reflecting the singularity exponent maintain well across different levels, but the scaling relation representing local fractal dimension may break due to the $q$ values are too high or too low. This suggests that fractal dimension is the strictest scaling exponent, and the scaling development difference between the singularity exponent and local fractal dimension can be employed to disclose the fractal structure evolution stages.

There are still many areas for improvement of our research in the future. The shortcomings of this study are as below: First, our data only covered Chinese cities. Further works might take traffic networks in other countries into account. Second, we only considered the spatial distribution of traffic networks. The complexity of a traffic network is related not only to its geometric form, but also to the dynamic process. Further analysis should be performed for the dynamic evolution of traffic networks. Third, multifractal description of traffic network is not visual enough. In other words, the description and explanation of fractal parameters for urban street networks are too abstract to be understood by beginners. How to explain the results of multifractal description in simple terms is one of the directions in the future. Fourth, the algorithm of parameter estimation was limited to the ordinary least squares (OLS) method. The alternative algorithm of OLS is the maximum likelihood method (MLM). MLM is regarded as a better approach to estimating power exponent values. However, the effectiveness of MLM depends on the joint normal distribution of random variables. Whether or not the linear size of boxes and corresponding measurements satisfy joint Gaussian distribution is unclear for the time being. The algorithm based on OLS has a significant advantage, that is, it is good for slope estimation. Where power law relation is concerned, the slope on a log-log plot represents fractal dimension or scaling exponent. In practice, both the OLS method and MLM can be employed to calculate scaling exponents. If the power law relation is well developed, the two algorithms will give similar calculation values. In contrast, if the power law is not well developed, the OLS can give approximate calculation values, while the MLM gives abnormal calculation values. This suggests that OLS can be used to make an approximate estimation of fractal

dimension and scaling exponents, while the MLM can be utilized to distinguish power laws from fake power laws [75]. In short, each method has its advantages and disadvantages. The algorithm of model parameter estimation can be selected according to the research objective and the characteristics of the research object. In this sense, MLM can be utilized to identify the disordered structure of multifractal traffic networks. We have tested the regression estimation by using MLM in R, and calculated the multifractal parameters. In our cases, the two algorithms lead to the similar calculated results. For example, for Beijing, the values of generalized correlation dimensions $D_q$ and local dimension $f(q)$ based on OLS are highly consistent with the those based on MLM. As for the singularity exponents $\alpha(q)$, there are two outliers when $q$ = -4 and -6. But the two outliers have no significant influence to the analytical conclusions (S4 Table). In short, the results show that there is no significant differences between the OLS-based multifractal spectrums and those based on MLM. The consistency between MLM-based results and the OLS-based results suggest that the spatial structure of urban street networks well follows power laws.

## 5 Conclusions

In this work, multifractal theory is applied to characterizing the urban street network of twelve representative megacities in China. Based on the results and findings shown above, the main conclusions can be drawn as follows. **First, the urban street network in Chinese megacities universally displays multifractal structure**. The judgment basis lies in typical multifractal spectral curves. The generalized correlation dimension spectrums take on inverse S-shaped curves, and the local fractal dimension spectrums take on unimodal curves. This suggests the street network is a complex hierarchy system, with the basic feature of spatial heterogeneity. It is not enough to describe it through a single scaling process. Multifractal scaling analysis provides an alternative approach for characterizing the complex structure of traffic networks. **Second, the street networks of megacities in China are characterized by disorderly development in the fringe zones and sparse areas, and degraded fractal structure in central areas and high-density areas**. If the street links in fringe areas are disorderly distributed, the generalized correlation dimension will seriously exceed the Euclidean dimension of the embedded space. This goes beyond theoretical expectations. On the other hand, the local fractal dimension is very sensitive. When moment order becomes too high or too low, the goodness of fit of the local dimension becomes very low. This implies the degradation of fractal structure. The reason may be that the spatial pattern of urban street network in the central area tends to be relatively compact and saturated, leaving limited free space. In contrast, the hierarchical structure of traffic networks has not been well developed in suburban areas. **Third, scaling precedes the local fractal structure of street network, and the fractal structure in some parts of the city is not significant enough**. Though the urban street networks follow scaling law both from the global and local level, the local fractal dimensions display partial degradation when reaching a certain level. This illustrates that the fractal dimension is the strictest scaling exponent. With the local fractal dimensions, we can diagnose the specific level where the structural problem of traffic network development occurs. **Fourth, the spatial pattern of urban street networks of megacities in China displays the characteristics of multifractal aggregation, but with a potential diffusion tendency**. The left tails of the local fractal dimension spectrums are higher than the right tails, and the left ends of the singularity spectrums are lower than the right ends. This is an indication of spatial aggregation. However, the singularity spectrums incline to the left, implying the signs of spatial diffusion. This may suggest some contradictory factors in the evolution of traffic networks of Chinese cities, or the spatial development

of traffic networks is in a certain transformation period. The specific reasons need further research in the future.

## Supporting information

**S1 Appendix. The results of multifractal computation conducted by street nodes.**
(DOCX)

**S1 Table. Datasets of box-counting numbers for multifractal dimension estimation.** This CSV file contains the box-counting numbers of urban street networks for Beijing. From the results, 'FID_$\varepsilon = k$' is the unique ID of boxes under the linear scale $\varepsilon_k$ (e.g., when $k = 1$, there is only one box. And when $k = 2$, there are four boxes.). 'Number' represents the total length of street links falling into the corresponding box, related to $L_i(\varepsilon_k)$. Empty boxes are not counted. Using these results, we can calculate the multifractal parameters of urban street networks.
(CSV)

**S2 Table. Calculated process datasets for the OLS estimation.** This MS Excel file contains the calculated Renyi entropy and related variables for estimating global and local multifractal parameters, as mentioned in S1 Appendix.
(XLSX)

**S3 Table. Calculated results of global and local multifractal parameters based on OLS.** This MS Excel file shows our calculated results of multifractal parameters, including global parameters, generalized correlation dimension $D_q$ and mass exponent $\tau(q)$, and local parameters, singularity exponent $\alpha(q)$ and local fractal dimension $f(q)$. The goodness of fit ($R^2$), the $p$-value, and the standard error of each parameter are given.
(XLSX)

**S4 Table. The multifractal parameters of Beijing's street network estimated by the OLS and MLM.** This MS Excel file shows the calculated results of multifractal parameters, including generalized correlation dimension $D_q$, singularity exponent $\alpha(q)$, and local fractal dimension $f(q)$. The goodness of fit ($R^2$), the $p$-value, and the standard error of each parameter are given. The mass exponent $\tau(q)$ can be derived from $D_q$ by Eq (3).
(XLSX)

## Acknowledgments

The data support from "Geographic Data Sharing Infrastructure, College of Urban and Environmental Sciences, Peking University (http://geodata.pku.edu.cn) ". The support is gratefully acknowledged. The authors will thank Dr. Linshan Huang for enlightening discussions about this research. We also thank our Academic Editor and four anonymous reviewers whose constructive comments are helpful to improve the quality of this manuscript.

## Author Contributions

**Conceptualization:** Yanguang Chen.

**Data curation:** Yuqing Long.

**Methodology:** Yanguang Chen.

**Supervision:** Yanguang Chen.

**Writing – original draft:** Yuqing Long.

Writing – review & editing: Yanguang Chen.

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
