## [Decision Letter · Decision Letter 0]

11 Jun 2020

PONE-D-20-11344

Multifractal scaling analyses of urban street networks: the cases of twelve megacities in China

PLOS ONE

Dear Dr. Chen,

Thank you for submitting your manuscript to PLOS ONE. We invite you to submit a revised version of the manuscript that addresses the points raised during the review process.

We look forward to receiving your revised manuscript.

Kind regards,

Nan Zheng

Academic Editor

PLOS ONE

Additional Editor Comments:

The majority of the reviewers is quite negative towards this work. I would give a second chance for the paper, as the paper involves empirical efforts which have its value. Please read carefully the comments by the reviewers.

Journal Requirements:

3. Please update your data availability statement to detail how the Chinese digital navigation map data can be accessed by future researchers, for example by including a literature citation, URL link to the data, or contact details of the organisation which owns the data.

"This research was sponsored by the National Natural Science Foundation of China (Grant No.

41671167). The support is gratefully acknowledged."

" The funders had no role in study design, data collection and analysis, decision to publish, or preparation of the manuscript."

5. We note that Figures 3, 4 and B1 in your submission contain map images which may be copyrighted. All PLOS content is published under the Creative Commons Attribution License (CC BY 4.0), which means that the manuscript, images, and Supporting Information files will be freely available online, and any third party is permitted to access, download, copy, distribute, and use these materials in any way, even commercially, with proper attribution. For these reasons, we cannot publish previously copyrighted maps or satellite images created using proprietary data, such as Google software (Google Maps, Street View, and Earth). For more information, see our copyright guidelines: http://journals.plos.org/plosone/s/licenses-and-copyright.

5.1.    You may seek permission from the original copyright holder of Figures 3, 4 and B1 to publish the content specifically under the CC BY 4.0 license.

5.2.    If you are unable to obtain permission from the original copyright holder to publish these figures under the CC BY 4.0 license or if the copyright holder’s requirements are incompatible with the CC BY 4.0 license, please either i) remove the figure or ii) supply a replacement figure that complies with the CC BY 4.0 license. Please check copyright information on all replacement figures and update the figure caption with source information. If applicable, please specify in the figure caption text when a figure is similar but not identical to the original image and is therefore for illustrative purposes only.

6. Please ensure that you refer to Figure 2 in your text as, if accepted, production will need this reference to link the reader to the figure.

Reviewers' comments:

Reviewer's Responses to Questions

**Comments to the Author**

1. Is the manuscript technically sound, and do the data support the conclusions?

Reviewer #1: Yes

Reviewer #2: Yes

Reviewer #3: Partly

Reviewer #4: Yes

2. Has the statistical analysis been performed appropriately and rigorously? 

Reviewer #1: Yes

Reviewer #2: Yes

Reviewer #3: N/A

Reviewer #4: No

3. Have the authors made all data underlying the findings in their manuscript fully available?

Reviewer #1: Yes

Reviewer #2: Yes

Reviewer #3: Yes

Reviewer #4: Yes

4. Is the manuscript presented in an intelligible fashion and written in standard English?

Reviewer #1: Yes

Reviewer #2: No

Reviewer #3: No

Reviewer #4: Yes

5. Review Comments to the Author

Reviewer #1: The authors propose an interesting method to investigate the multifractal nature of traffic networks and they support their analysis with the example of the 12 most representative cities in China. They show the multi-dimensionality of the fractal space in contrast with the monofractal scale of urban network, when we consider the street nodes as element of the network (preferred respect to the link lines). They highlight the similarities and the divergence among the dataset of 12 cities, trying to explain the causes and the main characteristics.

In my opinion, the results and the methods proposed deserve certainly to be published and the choice of PLOS ONE seems to me the most accurate and appropriate for the journal.

Even if the English used in the paper is correct and in some part (last) also enjoyable, I found an important lack of coherence and order in the formulas and definitions, especially in all the first half of the paper. I warmly suggest to review this part and improve the readability and the clearness.

Here below some remarks:

1) The abstract results too technical and obscure, without proper definition of the cited parameters. It is fine to cite some statistical features but with a minimum of mathematical coherence.

2) All section 2 must be reorganized in a better way. Whenever the author introduce a feature (q, D_q, tau_q definition of multi and mono fractal) is SHOULD be followed but a proper definition. I found some of them only at the end of the section that clarified the parameters. Please, move the AT THE FIRST and EVERY time a new variable appears in the text.

3) Eq.5 N_i(alpha,epsilon) in the equation but right after the subscript 'i' disappears. Please, report what it stands for right before or after the eq. 5.

4) Fig. 1 ) It might help to put the value of the entropy of the example just to give an idea of the magnitude of different pattern

5) Top pg. 6. What their definition of 'interaction'?

6) pg. 7 ) There are some bullets in blue. The author might consider to write them in bold if they want to highlight those sentences.

7) What correspond a(0) and f(a(0))... is it q = 0? alpha doesn't appear in the text as function of something. At least not directly. Please, specify.

8) pg. 8 ) " reflecting by", please consider the correctness of this sentence.

9)Fig. 4 It results shifted and not well centered in the page. The label of the first row of figure are in the other page. Please, consider to reorder them and report the figure in only one page.

10) Fig. 7. The ticks of all axes are too distanced. please consider to put minor ticks and ticksLabel as well. Moreover the axes label are not clear to me (maybe badly centered?). In addition, the grey points present in all plots are not clear what do they represent. Please consider to put a legend and/or a clearer explanation. But, in general I am not convinced by the goodness of this linear fitting of the scatter plot, starting always for the origin. Do the authors considered to compute the critical q when the value in y-axis go from 0 to >0 ?

Overall, I suggest this paper for publication but I expect some improvement of the form.

Reviewer #2: The article proposed the method to test the urban network in 12 megacities in China to see whether the multifractal structure can be observed. Then the multifractal analysis is applied to gain the spatial characteristics of the urban network. The results indicate that the street networks have significant multifractal structures and also has the spatial heterogeneity and hierarchical structure.

From the paper, the following comments are provided:

• The article has new data and used in a well-structured method, but there is not much clarification of the contribution which needs to elaborate further. In the conclusion part, it says by knowing the multifractal structure of the urban network, the accessibility of the urban system can be improved via the optimization, but how it would be guided through this proposed method and the contribution of this article needs to be specified.

• Page 8, line 17: “Nevertheless, compared with street lines, the distribution of street nodes has been more stable over times.” The first thing is ‘times’ should be ‘time’ here. Some references needed to support this statement. I realize this issue has been re-raised in the Discussion part, so a general suggestion is to reframe this part to show a more precise illustration.

• Page 17: the discussion about the population distribution seems irrelevant to this part.

• Page 20: “The single scaling process leads to mono-fractal structure, showing a uniform distribution. While multiple scaling process lead to multifractal structure.” This statement is confusing and irrelevant to the first part of the conclusion. I inferred that whether the system is monofractal or multifractal merely based on model selection. Thus, it weeks the conclusion of the multifractal would be better to suit the real-world complexity.

• The phrase ‘asymmetric cascade structure’ can only be found in abstract while no other place in this article has mentioned this concept.

• The paper has too many minor grammar errors: missing articles; the plural form is not consistency, like Page 3 line 10: ‘has’ should be ‘have,’ line 11 ‘method’ is not agree in number with other words; Page 3 line 8: ‘meterials’ should be ‘materials,’ follows the ‘founctional’ do you mean ‘functional’? I urge another round of proofreading is necessary.

• Some sentences causing frustrations need to reconsider:

Page 1,line 4: “cities, systems of cities, and traffic networks” seems not parallel concepts;

Page 8,line 1: “twelve Chinese megacities in China…” is redundant

Page 8, line 5: “As these cities…” this sentence lack logic.

Reviewer #3: The paper conducts an analysis street networks in 12 mega-cities in China, based on the assumption that they are multifractal systems. Overall, considering the conclusions of the study, the idea seems to be an extreme over-complication of the problem at hand while the framework has major weaknesses. I strongly believe authors have failed to establish the necessity and rigorousness of their analysis over widely-used network analyses.

Major comments:

The paper is not well-written and there is an abundance of errors and hastily written sentences. The second paragraph in Introduction has a sum of more than a dozen grammatical and spelling and writing issues.

Many sentences in the text are very ambiguous and convey no specific message and need to be properly explained or removed from the paper. Examples include but they are not limited to those below, taken from the beginning of the paper:

- It is difficult to model and analyze complex networks by conventional mathematical methods.

- The radial dimension proved to be a special spatial correlation dimension of fractals.

- Multifractal method bears analogy with telescopes and microscopes in geographical spatial analysis.

A major weakness of the paper is its inability to deliver a clear and effortless description of the framework. Also, procedures such as “city clustering algorithm” and “head-tail breaks” are not described in the paper; especially the former is not at all widely-used and needs clear explanation.

There is a need for at least present a synthetic example of how a non-fractal network system would compare to those analysed in this paper. Perhaps by generating a network with insignificant fractality and then performing comparisons by means of D0, D1, and D2.

Significance of the parameters need to be tested, for example via determining the same parameters in well-studied structures with high, moderate, or no fractality.

A significant issue is what appears to be, ignoring the links (representing road segments) and focusing on the network nodes, when analysing the road networks. This can be immensely misleading, especially in a study like this where the goal is (at least to great extent) to characterize the network connectivity.

Density of nodes which determines the parameters found this study, has actually little to do with local connectivity. A large number of connected nodes with few connections make a sparse network. This is the reason that network density (as a connectivity index) is measured by the number of network links.

Minor comments:

From the paper: “If an urban system bears multifractal structure, then different subareas have different growth probabilities and density distribution.”

Based on the above statement, can we conclude non-multifractal structure indicate similar growth and density distributions? If yes, which I believe is incorrect, then authors should explain why. And if not, then there always disadvantage in using a complex analysis to characterize a simple property, and authors better provide why their framework is any different.

Try to reduce the usage of “;” when it is more appropriate to break the sentence in two; in many cases it is used inappropriately.

Although one might be able to guess based on the context, it is unclear what “if D0>D1>D2 significantly” means.

It is hard to understand how this sentence helps the logical flow in interpreting the fractality of urban traffic networks: “Where there is interaction in a geographical region, there is an urban system, and where

there is an urban system, there is a traffic network.”

The goal is really obscured when sections 2.1 is presented before 2.2. I suggest authors mix the two somehow, or try to involve more interpretation when presenting the abstract ideas of multifractality.

Figure 2 is never pointed out in the text. I think the reference to Fig. 3 in page 7 needs correction.

I believe authors should not write sentences in “bold,” in order to highlight their importance. Try to highlight the important points using appropriate and effective writing.

“Good order” as opposed to “disorder” should be explained clearly in concept and how it can be interpreted from the planner or user perspective. The way these abstract concepts appears in the paper, is not informative to the reader and only appears as unnecessary jargon here and there, only making the text hard to follow.

Reviewer #4: This paper investigates the multifractal scaling laws of urban traffic networks in 12 large Chinese cities. The study uses empirical data to show the existence of such laws. The paper is well-written. I have the following issues with the paper:

1. An important aspect in my opinion is that the findings should be related to some intuition and implications. In its current form the manuscript lack both. I am not sure what the intuition of the three dimensions is for the traffic network, traffic performance, etc. What do I learn from this analysis from D1, D2, and D3? Their three points in the conclusions are a good start, but I am not sure they are completely translated into traffic.

2. Similarly, the objective of the study is unclear to me. I believe that clearly stating the research gap and the objective would be emphasize the contributions of the work.

3. Sections Discussion and Conclusions are very redundant. I advise the authors to merge these sections.

4. The wording in the manuscript needs substantial revision for consistency. For example, D2 is referred as the correlation dimension; later it’s the connectivity (p10). There are many more examples, where terms are mixed and used interchangeably. I also advise the authors to use commonly used terms such as polycentric, monocentric etc. The term centripetal seems made up in this context. I assume street lines refer to links in a network. Please use the commonly used terms from network science, edges and vertices, or links and nodes.

5. Introductory figures 1 and 2 are a good idea, but they should also be put into the context of traffic networks. Could the authors give an example?

6. The section 2.1 and 2.2 are good, but they lack intuition (see also my 1st comment). It would help if this would be translated to traffic and traffic networks more rigorously.

7. The statistical analysis is not convincing to me. Showing an R-squared for a fixed intercept is not a good idea, as it loses its interpretation. No p-values are shown.

8. Similarly, estimating a power law with a simple OLS can be cumbersome. I advise to test their exponents with common packages as poweRlaw in R or similar in python. These take into accounts many errors that happen generally when using OLS directly.

9. The study is limited to Chinese cities, this is not a drawback, but should be mentioned as a limitation.

10. This might be out of scope, but I advise the author to check the sensitivity to which the CCA is applied. An easy approach might be to just take the city center and consider a varying radius and see the influence thereof.

6. PLOS authors have the option to publish the peer review history of their article (what does this mean?). If published, this will include your full peer review and any attached files.

Reviewer #1: No

Reviewer #2: No

Reviewer #3: No

Reviewer #4: No

---

## [Author Response · Author response to Decision Letter 0]

18 Aug 2020

See the attached file titled "Response to Reviewers".

---

## [Decision Letter · Decision Letter 1]

24 Nov 2020

PONE-D-20-11344R1

Multifractal scaling analyses of urban street network structure: the cases of twelve megacities in China

PLOS ONE

Dear Dr. Chen,

Thank you for submitting your manuscript to PLOS ONE. After careful consideration, we feel that it has merit but does not fully meet PLOS ONE’s publication criteria as it currently stands. Therefore, we invite you to submit a revised version of the manuscript that addresses the points raised during the review process.

ACADEMIC EDITOR:

Thanks for your resubmission. Please address the remaining comments by two reviewers. 

We look forward to receiving your revised manuscript.

Kind regards,

Nan Zheng

Academic Editor

PLOS ONE

Additional Editor Comments (if provided):

Two reviewers are still very critical on the paper. One wrote me the comments directly, as there was some account issues. I copy-paster the comments below. Please address both reviewers.

"Although authors have applied my minor comments and one of my major comments regarding the network links, yet in my opinion the paper is not sufficiently well-written to be useful for its target audience.

Authors have started their response by providing clarification on necessity and rigor of their study. The explanation (similar to the manuscript) contains too many grammatically incorrect sentences such as “Multifractal theory does be an excellent mathematical tool…” which makes the explanations hard to follow.

In my opinion, authors misunderstood the comment about “over-complication of the problem at hand.” This means that simpler methods can deliver even a clearer and easier-to-interpret analysis of street networks.

One of my concerns was ambiguity of arguments in this manuscript. Now in their response authors state that “Through the spectral curves, we can examine the different parts and different levels of a geographical system.” This is another example of ambiguity. What are “different levels of a geographical system”? There are numerous attributes that exist at different levels in a geographical system. If by this the authors mean, say, “different levels of granularity,” then they should write so in a clear and understandable manner. A publishable manuscript needs to contain minimum occurrence of such sentences.

Again, in comment 1 I quoted just a few sentences as examples which in my opinion do not explain the technical issues in a proper manner, e.g. “It is difficult to model and analyze complex networks by conventional mathematical methods.” Authors have responded by “Dear reviewer, all the following statements are easy to understand if you are familiar with multifractal theory.” This shows an issue with authors’ approach to technical writing; no matter the target audience, technical writing needs to be sound and clear. The problem with the quoted sentence has nothing to do with “familiarity with the topic,” i) you are making a wrong statement about conventional mathematical methods, which indeed are able to model and analyze complex networks, and ii) fractal analysis can also be deemed as a conventional mathematical method, but you are painting it differently as it is an advantage!

Authors refused to apply comment 3 and responded “Nonfractal systems have no fractal dimension.” This is just not true. A non-fractal system tends to have a flatter D_(Q) vs Q. This is one of the fundamental aspects of fractal analysis."

Reviewers' comments:

Reviewer's Responses to Questions

**Comments to the Author**

1. If the authors have adequately addressed your comments raised in a previous round of review and you feel that this manuscript is now acceptable for publication, you may indicate that here to bypass the “Comments to the Author” section, enter your conflict of interest statement in the “Confidential to Editor” section, and submit your "Accept" recommendation.

Reviewer #1: All comments have been addressed

Reviewer #2: All comments have been addressed

Reviewer #4: All comments have been addressed

2. Is the manuscript technically sound, and do the data support the conclusions?

Reviewer #1: Partly

Reviewer #2: Yes

Reviewer #4: Partly

3. Has the statistical analysis been performed appropriately and rigorously? 

Reviewer #1: Yes

Reviewer #2: I Don't Know

Reviewer #4: No

4. Have the authors made all data underlying the findings in their manuscript fully available?

Reviewer #1: Yes

Reviewer #2: Yes

Reviewer #4: Yes

5. Is the manuscript presented in an intelligible fashion and written in standard English?

Reviewer #1: Yes

Reviewer #2: Yes

Reviewer #4: Yes

6. Review Comments to the Author

Reviewer #1: I consider that the authors replied adequately to my comments. Even if I still have some reservations about the interpretation of the results given by the authors, I consider the paper at a good level for publication. Just a little remark: Fig. 1 should be placed on the same page than its caption. Please, consider figure and its caption as a unique object.

Reviewer #2: (No Response)

Reviewer #4: I thank the authors for the revision of their manuscript. In the course of this evaluation, I have also read the reviews of the other reviewers. Interestingly, the third reviewer raises similar issues, especially about the over-complication. As a general comment, I still lack the connection to traffic itself. Table 1 is a good example of how the study still fails to demonstrate to show this connection.

I believe that the authors have not addressed all of my comments appropriately. There are also multiple careless spelling errors in the revision letter. I still have open questions about the following responses.

R1: The table does still not relate to traffic.

R2: The aim is still unclear. What are the implications of the work, what do I learn from it? It still seems to be a research for the sake of research.

R6: I am somewhat doubtful about the OLS answer. I advise the authors to compare their results with the MLM approach.

7. PLOS authors have the option to publish the peer review history of their article (what does this mean?). If published, this will include your full peer review and any attached files.

Reviewer #1: No

Reviewer #2: No

Reviewer #4: No

---

## [Editor Report · Decision Letter 2]

20 Jan 2021

PONE-D-20-11344R2

Multifractal scaling analyses of urban street network structure: the cases of twelve megacities in China

PLOS ONE

Dear Dr. Chen,

Thank you for submitting your manuscript to PLOS ONE. After careful consideration, we feel that it has merit but does not fully meet PLOS ONE’s publication criteria as it currently stands. Therefore, we invite you to submit a revised version of the manuscript.

Given that two reviewers were positive and one reviewer was quite negative, I decided to have a look at the response letter myself. It looks like you have made your efforts, but Reviewers 3 and 4 did raise several good points. So my decision is this. I would not send it again for review and conditionally accept the paper, if you agree on (1) at least make 1~2 changes suggested by reviewer 4, and (2) clearly acknowledge the limitations raised by the two reviewers.  

We look forward to receiving your revised manuscript.

Kind regards,

Nan Zheng

Academic Editor

PLOS ONE

---

## [Editor Report · Decision Letter 3]

29 Jan 2021

Multifractal scaling analyses of urban street network structure: the cases of twelve megacities in China

PONE-D-20-11344R3

Dear Dr. Chen,

We’re pleased to inform you that your manuscript has been judged scientifically suitable for publication and will be formally accepted for publication once it meets all outstanding technical requirements.

Kind regards,

Nan Zheng

Academic Editor

PLOS ONE

Additional Editor Comments (optional):

Final comment, consider add the following papers as they are relevant and innovative on the same subject of study:

Understanding traffic capacity of urban networks A Loder, L Ambühl, M Menendez, KW Axhausen, Scientific reports 9 (1), 1-10

Analysis of one-way and two-way street configurations on urban grid networks, J Ortigosa, VV Gayah, M Menendez, Transportmetrica B: transport dynamics

Analysis of network exit functions for various urban grid network configurations, J Ortigosa, M Menendez, VV Gayah, Transportation Research Record 2491 (1), 12-21
---

## [Editor Report · Acceptance letter]

3 Feb 2021

PONE-D-20-11344R3 

Multifractal scaling analyses of urban street network structure: the cases of twelve megacities in China 

Dear Dr. Chen:

I'm pleased to inform you that your manuscript has been deemed suitable for publication in PLOS ONE. Congratulations! Your manuscript is now with our production department. 

Kind regards, 

on behalf of

Dr. Nan Zheng 

Academic Editor

PLOS ONE